# A study on the impact of tourism destination image and local attachment on the revisit intention: The moderating effect of perceived risk

**Jiahua Wei**[ID]*, **Lewei Zhou, Lei Li***

Business School, Guilin University of Technology, Guilin, Guangxi, China

* jiahua6688@163.com (JW); lileiguilin@foxmail.com (LL)

## Abstract

The revisit intention of tourists is an important guarantee for the sustainable and healthy development of tourism destination, and has also received attention from the current academic community. However, there is still insufficient research on the antecedents of revisit intention from the perspectives of tourism destination, image and nostalgia emotion. This study takes China's ecological tourism scenic area (Guilin Lijiang Scenic Area) as a case study, and uses questionnaire survey method to obtain research data for empirical research. The results of this study confirm that tourism destination image has a positive impact on nostalgia emotions and local attachment, nostalgia emotion has a positive impact on local attachment, and local attachment has a positive impact on revisit intention. Perceived risk plays a negative moderating effect between local attachment and revisit intention. In addition, this study also examined the mediating effect of nostalgia emotion and local attachment. This study is beneficial for enriching the theory of the influence mechanism of revisit intention from the perspective of consumer psychology. It is an interdisciplinary research result of management and psychology, providing theoretical reference for improving revisit intention in tourism destinations and promoting their healthy development.

## 1. Introduction

In December 2022, the Chinese government released the COVID-19 prevention and control measures, marking the end of China's three-year strict "dynamic zero" prevention and control policy. In this context, China's tourism industry has experienced rapid recovery. During the May Day holiday period in China in 2023 (from April 30, 2023 to May 3, 2023), the total number of domestic tourism trips in China was 274 million, a year-on-year increase of 70.83%, and recovered to 119.09% of the same period in 2019 according to comparable standards. However, in the process of recovering the tourism industry, some tourism destinations have also experienced some tourism chaos, mainly reflected in illegal tour guides, unreasonable low-price groups, opaque restaurant food prices, tour guides insulting customers, and products

**Data Availability Statement:** All relevant data are within the manuscript and its Supporting Information files.

**Funding:** 1. Guangxi Natural Science Foundation (Grant No.2021GXNSFAA220066). 2. National Natural Science Foundation of China (Grant No. 72262009). 3.National Natural Science Foundation of China (Grant No. 72074058).

**Competing interests:** The authors have declared that no competing interests exist.

being inferior, and so on. These phenomena have led to a large number of tourist complaints, seriously affecting tourist satisfaction and revisit intention, and affecting the brand image and economic benefits of tourism destinations. Therefore, how to enhance tourism destination image, enhance tourists' nostalgia emotion and local attachment, reduce tourists' perceived risks, and thus increase tourists' revisit intention has become an urgent and important issue that needs to be addressed by tourism destinations and the theoretical community.

In current tourism research, the impact of destination image on post purchase behavior and tourist behavioral intention has been studied [1]. Fan et al. believe that several elements of cognitive imagery in tourism destinations will have a direct impact on emotional image, which in turn will directly affect local attachment [2]. However, the direct impact of tourist destination image on local attachment is rarely addressed. Some scholars conducted empirical research on the relationship between local attachment and willingness to revisit [3, 4]. However, the moderating effect of local attachment on revisit intention is still rarely addressed in current literature. And under the specific situation of China's liberalization of the prevention and control of the COVID-19, how to interpret the influence mechanism between the tourism destination image, nostalgia emotion, local attachment and revisit intention? How does perceived risk have a moderating effect? This is a new topic for the academic community, and there is currently a lack of systematic research, which requires further theoretical exploration.

This study is based on two aspects of theoretical exploration and tourism management practice, and takes Guilin Lijiang Scenic Area as a case study to conduct empirical research. In terms of research methods, this study collected research data through questionnaire survey, and used SPSS 26.0 and AMOS 26.0 to conduct descriptive analysis, reliability and validity analysis, multiple regression analysis, etc. to deeply analyze the impact of tourism destination image on nostalgia emotion and local attachment, the impact of nostalgia emotion on local attachment, and the impact of local attachment on revisit intention. On this basis, this study will also examine the moderating effect of perceived risk between local attachment and revisit intention, as well as the mediating effect of nostalgia emotion and local attachment. This study is conducive to in-depth exploration of tourists' revisit intention from the perspectives of psychology and management, and is a new attempt at interdisciplinary research. This study will enrich the research theory of consumer psychology, expand the research perspective of tourism management, provide theoretical reference and practical guidance for tourism destinations and tourism enterprises to improve their brand image and revisit intention, and promote the healthy development of China and even the world's tourism industry.

## 2. Literature review and research hypotheses

### 2.1. Tourism destination image

The "image" originally refers to the unique sensory image of a city to its residents, which is a "common psychological image" based on visible entities [5]. Tourism destination image is a complex and dynamic interactive system, whose core lies in the cognitive and emotional reactions to the common and unique attributes of the destination. It is a combination of tourists' beliefs, thoughts, feelings, expectations, and impressions [6, 7]. The current literature on tourism destination image is very extensive, including its concept, measurement, factors, and the mediating role of tourism destination image on tourist intention, behavior, and satisfaction [8] Recent researchers have used social network methods to analyze the distribution of tourism destination image, suggesting that the distribution of tourism destination images follows a "core-edge" pattern [9]. Tourism destination image not only has a significant impact on tourists' tourism decisions and subsequent behaviors, but also has a close relationship with the competitiveness of tourism destinations [10]. Based on the theory of "cognition emotion", the

academic community has condensed four dimensions of tourism destination image: Landscape image, cultural image, local image, and emotional image [11]. A good tourist experience can receive positive reviews from tourists, while a bad tourist experience can receive negative reviews from tourists [12]. The main impact factors of tourism destination image include six factors: Core attraction, communication, expectation, behavior, service, and local projection [13]. These factors will greatly affect tourists' perception of this tourism destination, thereby affecting their decision-making behavior. Tourism destination image is the result of the interaction between tourists and the destination. Through participation, tourists invest their time and effort, interact with the landscape, facilities, and services of the destination, forming an overall perception of the destination [13]. Several elements of cognitive image in tourism destinations directly impacts emotional image, which in turn directly impacts local attachment and indirectly impacts tourists' environmental responsibility behavior through the variable of local attachment [2]. However, there are still many deficiencies in the research on tourism destination image in the academic community, and there is still a lack of research on the impact of tourism destination image on tourists' revisit intention, which requires in-depth research by the academic community. This study uses tourism destination image and other psychological variables as antecedent variables to explore their impact mechanisms on revisit intention, expanding the research perspective in this field.

## 2.2. Nostalgic emotion

According to psychological explanations, nostalgia is a psychological defense mechanism, in which individuals tend to seek a sense of security through nostalgia when encountering contradictions or conflicts [14]. Nostalgia emotion is feedback of emotional generated by individuals about things that have happened before, usually accompanied by feelings of bitterness or sweetness [15]. At the social level, nostalgia emotion can be divided into two types: Individual nostalgia and collective nostalgia [16]. Stern divides nostalgia emotions into historical nostalgia and personal nostalgia from a cultural perspective, which is also the main classification of nostalgia emotion types [17]. In recent years, changes in tourist behavior have been seen as a driving force for emphasizing the importance of unpopular tourism destinations, which is one of the key potentials of any country's tourism industry. Unpopular tourism destinations still have the potential to be developed and bring more benefits to the local economy through comprehensive site selection methods [18]. Current research has found that tourists' nostalgia emotion has a positive impact on their tendency to travel [19]. The impact of nostalgia emotion on the variable of local attachment has internal differences. Nostalgia emotion has a significant impact on local dependence, but its impact on local identity is not significant [20]. Personal nostalgia emotion will have a significant positive impact on historical nostalgia emotion [21] Tourism destination image, as a key pre variable driving tourist behavior, has been a focus of previous studies on tourist choice intention, post purchase behavior, and behavioral intention [1]. The cognitive image of tourism destination image directly impacts subsequent emotional image, while nostalgia emotion, as a form of emotion, is impacts by the cognitive tourism destination image. Therefore, the research hypothesis H1 is proposed:

H1: Tourism destination image has a positive impact on nostalgia emotion.

## 2.3. Local attachment

The concept of local attachment has received widespread attention in the tourism industry. Although the concept of local attachment has not yet formed a completely unified definition, most scholars believe that local attachment reflects a type of human land relationship, which describes an emotional connection between an individual and factors such as their living

environment [22, 23]. Some scholars also believe that local attachment in tourism destinations is a connection between tourists and the destination based on emotions, cognition, and behavior [24]. Currently, most research on local attachment in China adopts the two-dimensional structure proposed by Williams [25], which analyzes local attachment from two aspects: Local dependence and local identity. Among them, local dependence belongs to a functional dependence of individuals, reflecting the relationship between the material conditions of tourism destinations and individual needs [26]. Local identity, on the other hand, is an emotional dependency that can give tourists a deeper sense of belonging to the destination [27]. Tourists will first develop a sense of dependence on a certain tourism destination after traveling, and only by deepening their connection with the destination after revisiting can they gradually form a sense of identity [28]. Huo et al. (2023) pointed out that local dependence can not only positively affect local identity, but also indirectly have a positive impact on local identity through the mediating effect of tourists' psychological ownership [29]. The dimension of local dependence has a significant positive impact on local identity, but research has shown that the correlation between the two is not high [30]. From the above literature, it can be seen that there is still relatively little research on the relationship between tourism destination image and local attachment in the tourism field. However, relevant literature has also explored the relationship between the two variables. For example, current research suggests that in sports events, the image of the hosting location has a significant positive impact on local dependence [31]. In addition, the recognition and involvement of tourism destination image play a promoting role in cultivating tourist attachment to the place [27]. Therefore, this study proposes the following research hypotheses:

H2: Tourism destination image has a positive impact on local attachment.

Nostalgia emotion can also stimulate positive emotional experiences, enhance positive evaluations, strengthen social connections, enhance a sense of belonging, and promote the formation of identity [14]. In the study of emotional relationships, Swets and Cox found a negative correlation between nostalgia preference and avoidance of attachment, and a positive correlation with relationship quality [32]. In the study of tourism scenarios, Jiang and Sun took the historical and cultural district of Liwan District, Guangzhou City as the research object, and used the method of structural equation model to analyze 346 valid questionnaires [33]. The research results showed that tourists' nostalgia emotion had a direct positive impact on local attachment, but also had an indirect effect through the mediating effect of perceived value. Therefore, this study proposes the following research hypotheses:

H3: Nostalgia emotion has a positive impact on local attachment.

## 2.4. Revisit intention

Customer loyalty refers to the preference and repetitive purchasing behavior of customers towards a company's products or services over a long period of time [34]. The revisit intention indicates a loyalty of tourists to the destination [35]. The willingness to revisit is a manifestation of consumers' sustained trust. Consumer sustained trust refers to the trust generated after the first transaction between consumers and businesses, which is divided into three dimensions: sustained ability, sustained integrity, and sustained goodwill [36]. Shankar et al. summarized three types of factors that affect online shopping trust, including consumer characteristics, merchant characteristics, and interaction factors [37].

Some scholars believe that revisit intention is similar to repurchase intention, which is a behavioral tendency of tourists to travel to a tourism destination again after developing a satisfactory emotion towards the tourism real estate [38]. Revisiting can promote the sustainable development of tourism destinations and enable them to achieve long-term development [39].

Abubakar et al. confirmed that EWOM positively impacts revisit intention and destination trust, while destination trust impact revisit intention [40]. Both tourism motivation and willingness to interact positively impacts tourists' revisit intention [41]. Based on meta-analysis, some scholars have proposed six factors that impact the revisit intention, including tourism motivation, past travel experiences, destination image, tourist satisfaction, perceived value, and perceived attractiveness [42]. Li et al. believe that tourism destination image has a significant positive impact on both tourist satisfaction and revisit intention, and tourist satisfaction also has a significant positive impact on revisit intention [43]. Therefore, the study of revisit intention needs to consider multiple factors, and only considering the impact of a single variable on it is not complete [44].

Data collection was conducted in six professional baseball games in South Korea, and 461 responses were collected from sports consumers. The study confirmed that consumers' local attachment has a positive impact on tourists' revisit intention [45]. Zhang et al. taking Shaoshan Scenic Spot in China as a case study, conducted semi-structured interviews with revisited tourists, and analyzed the data using the Grounded theory method [3]. The study confirmed that tourists' local attachment has a positive impact on revisit intention. Aghnia and Pratiwi conducted a questionnaire survey on 315 tourists to Taman Lapangan Banteng, Jakarta, Indonesia, and found that local attachment played a mediating impact between scenic spot satisfaction and revisit intention [4]. Therefore, based on the above analysis, this study proposes the following three research hypotheses:

H4: Local attachment has a positive impact on revisit intention.

H5: Local attachment plays a mediating effect in the impact of tourism destination image on revisit intention.

H6: Nostalgia emotion and local attachment play a mediating effect in the impact of tourism destination image on revisit intention.

## 2.5. Perceived risk

Perceived risk refers to the combination of various uncertainties and risks perceived by consumers during the consumption process [46]. Perceived risk is the subjective perception of consumers, and the magnitude of actual risk can only have an impact on their consumption behavior if it is subjectively perceived by consumers [47]. In terms of the dimensions of perceived risk, perceived risk can be divided into credit risk, delivery risk, appearance risk, physical risk, and purchasing method risk [48], as well as transaction risk, information risk, delivery risk, and after-sales risk [49]. Ren et al. confirmed that for consumers in online shopping, seller reputation, website construction, and product quality have a significant negative impact on the relevant dimensions of perceived risk [50]. Various dimensions of perceived risk (social risk, time risk, and financial risk) have a negative impact on online purchase intention [51]. Kim & Eishin conducted a survey and analysis on 558 consumers (409 Koreans and 149 Japanese) who had purchased fast food at convenience stores, and found that consumers' perceived risk would reduce their willingness to repurchase [52]. Xie et al. introduced the concept of tourist perceived risk in the context of forest health tourism, and found that the perceived risk of potential tourists has a significant negative impact on behavioral intention. In tourism scenarios, if the perceived risk of tourists increases, it will reduce the positive impact of local attachment on revisit intention [53]. Therefore, the following research hypotheses are proposed:

H7: Perceived risk plays a negative moderating role in the impact of local attachment on revisit intention.

Based on the above literature analysis and research assumptions, this study constructed a research model. As shown in Fig 1.

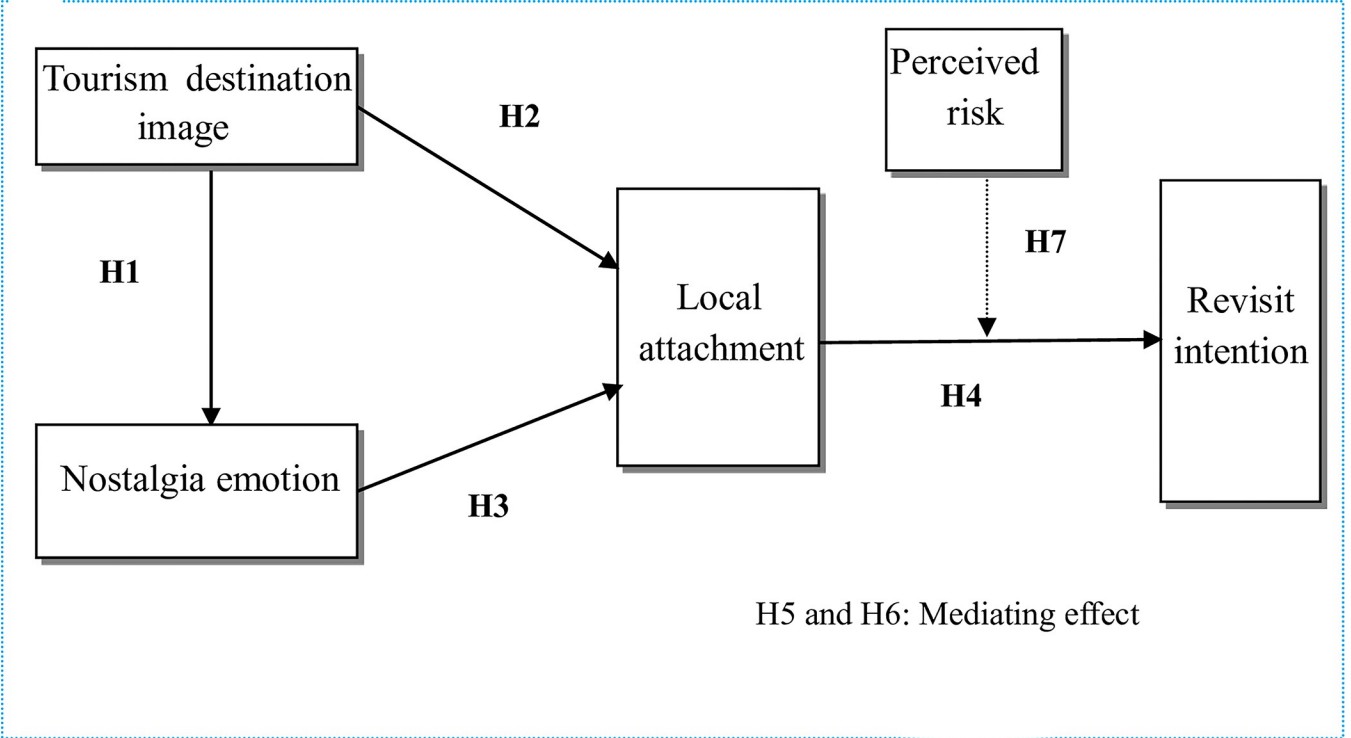

**Fig 1. Research model diagram.**

## 3. Research design

### 3.1. Selection of tourism destination cases

This study takes the Guilin Lijiang Scenic Area in China as a case study. Guilin Lijiang Scenic Area, as a case study, can well fit the research context of this article. Guilin is a famous tourism city in China, and the Guilin Lijiang Scenic Area is a world natural heritage site, which can provide sufficient representative samples for empirical research. Guilin Lijiang Scenic Area has many tourism resources, such as One River (Lijiang), Two Caves (Ludi Rock, Qixing Rock), and Three Mountains (Duxiu Peak, Fubo Mountain, and Diecai Mountain). The scattered nature of the scenic spots provides sufficient time for tourists to communicate with local residents and understand local culture. In addition, Guilin Lijiang Scenic Area has rich cultural resources, which can effectively shorten the psychological distance between residents and tourists, creating conditions for host guest communication. Guilin Lijiang Scenic Area attracts tourists from all over the world (over 90% are from China), which to some extent ensures the representativeness of the research sample in terms of space.

### 3.2. Variable measurement

This study involves five variables: Tourism destination image, nostalgia emotion, local attachment, perceived risk, and revisit intention. The measurement items of the five variables in this study are all sourced from authoritative measurement scales currently available. On this basis, researchers made necessary adjustments and modifications to the textual narrative style and some items of the scale through expert interviews and small sample testing based on the specific scenario of this study, thus forming a formal measurement scale. The sources of measurement items for the 5 variables are shown in Table 1.

**Table 1. Reference sources of measurement items.**

| Measurement variable | Reference source | Number of measurement items |
|---|---|---|
| Tourism destination image | Beerli & Martin (2004) [54], Fan et al. (2014) [2], Pan & Wang (2018) [55] | 6 |
| Nostalgia emotion | Verma & Rajendran (2017) [56], Yu et al. (2022) [20], Fan et al. (2022) [21] | 5 |
| Local attachment | Xu et al. (2022) [57], Qin (2022) [58], Ramkissoon (2013) [59] | 5 |
| Perceived risk | Li & Li (2017) [60], Lee (2015) [48] | 5 |
| Revisit intention | Luo & Yao (2018) [61], Wang et al. (2022) [62], Oppermann (2000) [63] | 3 |

All measurement items were measured using the Likert 5-point scale. During the questionnaire development process, researchers converted the items in the English scale into Chinese and then translated them into English through a standard translation back translation process. In order to further ensure the effectiveness of the questionnaire, researchers specifically consulted scholars and industry experts in the fields of tourism management and psychology on whether the questionnaire items were reasonable before conducting formal research. Based on this, the questionnaire items were adjusted and formed a formal questionnaire. The survey questionnaire also set measurement items such as gender, age, source, education, occupation, etc. to investigate the personal situation of the participants. Please refer to the appendix for the questionnaire.

### 3.3. Questionnaire survey

Due to the difficulty in obtaining data on tourism destination image, nostalgia emotion, local attachment, perceived risk, and revisit intention through public sources, this study used a questionnaire survey to collect data. The questionnaire survey was conducted from April 10, 2023 to May 7, 2023. This study obtained samples of survey questionnaires through the following channels: Firstly, cooperating with Guilin Lijiang Scenic Area and related hotels to obtain the contact information of tourists to Guilin Lijiang Scenic Area within a year, such as phone number, QQ, WeChat, etc., inviting them to participate in the survey, filling out the questionnaire online, and giving each person a cash incentive of 80 yuan. Secondly, researchers and volunteers participating in the study conducted on-site questionnaire surveys at multiple scenic spots in Guilin Lijiang Scenic Area. The questionnaires were collected on-site and each person was given a cash incentive of 50 yuan. The cash incentive funds come from the research projects supported by this study.

In the end, this study followed the opinions of Desimone et al. on questionnaire data processing [64], and screened based on the actual situation of the questionnaire. Five consecutive questions were removed and the same answer was chosen, resulting in inconsistencies and highly similar options. A total of 367 valid questionnaires were collected. The sample statistics are shown in Table 2.

## 4. Data analysis

### 4.1. Common method deviation testing

This study referred to the common method bias (CMB) test method proposed by Li & Zhang [65]. In order to reduce the impact of common method bias, in the implementation of the research, this study adopts a multi time point and multi-source approach to reduce common method bias (CMB). In addition, this study used SPSS 26.0 to conduct Harman univariate tests on five key variables, with a total variance explanatory power of 72.35% and a variance explanatory power of 30.14% for the first principal component, which is less than the 40% threshold.

**Table 2. Sample statistics.**

| One class indicators | Two class indicators | sample size | Percentage | Oneclass indicatos | Two class indicators | Sample size | Percentage |
|---|---|---|---|---|---|---|---|
| Gender | Male | 178 | 48.50% | Education | Secondary school and below | 117 | 31.88% |
| | Female | 189 | 51.50% | | College degree | 121 | 32.97% |
| Region | First-tier cities | 58 | 15.80% | | Bachelor | 98 | 26.70% |
| | Second-tier cities | 96 | 21.16% | | Master and doctor | 31 | 8.45% |
| | Third-tier cities and and other cities | 104 | 28.34% | Single travel time | Under 6 hours | 102 | 27.79% |
| | County and township | 75 | 10.44% | | 6–12 hours | 169 | 46.05% |
| | Rural area | 34 | 9.26% | | Over 12 hours | 96 | 26.16% |
| Age | Under 22 years old | 67 | 18.26% | Number of visits | 1 time | 111 | 30.25% |
| | 23-35years | 135 | 36.78% | | 2 times | 148 | 40.33% |
| | 36–59 years | 109 | 29.70% | | 3 times | 84 | 22.89% |
| | Over 60 years | 56 | 15.26% | | Over 3 times | 24 | 6.54% |

Data source: According to the questionnaire survey sample statistics of this study

Therefore, the above analysis confirms that the survey questionnaire data of this study passed the common method bias (CMB) test.

## 4.2. Reliability and validity testing

After obtaining research data through a questionnaire survey, this study conducted reliability and validity tests on the survey questionnaire data. In the reliability test, this study drew on the test standards and methods of Wu [66]. believing that the Cronbach's α should be greater than 0.70 to pass the reliability test. In Table 3, Cronbach's α range from 0.732 to 0.865, all greater than 0.70, meeting the reliability test criteria and therefore passing the reliability test.

As shown in Table 3, the factor load of all items is greater than the threshold value of 0.50, and the significance also meets the requirements, all are less than 0.05. The combined reliability (CR) of the five variables ranges from 0.858 to 0.918, which is greater than the threshold value of 0.70. In addition, the average variance extracted (AVE) of the five variables ranges from 0.590 to 0.692, which is greater than the threshold value of 0.50. Based on the above analysis data, the convergence validity has passed the test.

As shown in Table 4, among the five variables of tourism destination image, nostalgia emotion, local attachment, perceived risk and revisit intention, the data on the diagonal is the square root of the AVE for each variable. It can be seen that the correlation coefficients between each variable are smaller than the square root of the corresponding variable's AVE. Therefore, the discriminant validity test was passed. In addition, the structural validity test is shown in Table 5, and all fitting indicators of the model in this study meet the standards of a good model, confirming that the model in this study has good structural validity.

## 4.3. Research hypothesis testing

**4.3.1. Direct impact test.** To test the direct and moderating effects in this study, a multi-level regression analysis was conducted using SPSS26.0. Before the multi-level regression analysis, the researchers conducted a centralized analysis on each variable to avoid multi-collinearity. The variance inflation factor (VIF) of this study is between 3.447–5.289, which is lower than the empirical value of 10, indicating that there is no multi-collinearity problem in the research model. The multi-level regression results of this study are shown in Table 6.

As shown in Table 6, Model 1 is a regression of nostalgia emotion using control variables. The research results show that, except for the weak impact of age and single visit frequency on

**Table 3. Test of reliability and validity.**

| Variables | Measurement items | Load factor | T | Cronbach's α CR AVE |
|---|---|---|---|---|
| Tourism destination image | 1. Guilin Lijiang Scenic Area is very beautiful | 0.869 | 2.808 | 0.712 0.888 0.577 |
| | 2.Guilin Lijiang Scenic Area makes me Relax and Relax | 0.898 | 3.187 | |
| | 3. The equipment in Guilin Lijiang Scenic Area is very complete | 0.511 | 2.879 | |
| | 4.The service quality of Guilin Lijiang Scenic Area is high | 0.831 | 4.196 | |
| | 5. Guilin Lijiang Scenic Area is safe and reliable | 0.679 | 2.442 | |
| | 6.Guilin Lijiang Scenic Area is pleasant | 0.698 | 2.056 | |
| Nostalgia emotion | 7. The tourism experience in Guilin Lijiang Scenic Area has left me unforgettable | 0.735 | 2.349 | 0.711 0.866 0.565 |
| | 8.The tourism experience in Guilin Lijiang Scenic Area makes me feel happy | 0.709 | 2.835 | |
| | 9.The tourism experience in Guilin Lijiang Scenic Area reminds me of my loved ones | 0.731 | 3.289 | |
| | 10.The tourism experience in Guilin Lijiang Scenic Area has made me reminisce about my past life | 0.719 | 2.472 | |
| | 11.The tourism experience in Guilin Lijiang Scenic Area has given me the motivation to live | 0.855 | 3.071 | |
| Local attachment | 12.Guilin Lijiang Scenic Area has given me a sense of belonging | 0.832 | 2.713 | 0.875 0.890 0.626 |
| | 13.My heart still stays in Guilin Lijiang Scenic Area | 0.676 | 4.192 | |
| | 14. I like Guilin Lijiang Scenic Area | 0.920 | 4.234 | |
| | 15.I acknowledge the culture of Guilin Lijiang Scenic Area | 0.890 | 3.670 | |
| | 16.I acknowledge the services of Guilin Lijiang Scenic Area | 0.583 | 3.983 | |
| Perceived risk | 17. I'm worried that I don't have time to travel | 0.564 | 7.034 | 0.869 0.846 0.529 |
| | 18. The cost of Guilin Lijiang Scenic Area is too high | 0.842 | 5.862 | |
| | 19.The service quality of Guilin Lijiang Scenic Area is not high | 0.765 | 2.871 | |
| | 20.Incomplete safety measures in Guilin Lijiang Scenic Area | 0.619 | 3.943 | |
| | 21.The food in Guilin Lijiang Scenic Area is not what I like | 0.807 | 2.973 | |
| Revisit intention | 22. I am willing to revisit the Lijiang Scenic Area in Guilin | 0.852 | 3.872 | 0.752 0.855 0.663 |
| | 23. I am willing to revisit the newly developed scenic spots of Guilin Lijiang Scenic Area | 0.782 | 2.964 | |
| | 24.I will recommend friends to visit the Lijiang Scenic Area in Guilin | 0.807 | 2.541 | |

Data source: Statistics based on the questionnaire survey data of this study

nostalgia emotion, other control variables have no statistically significant impact on nostalgia emotion. Model 2 shows that tourism destination image has a significant positive impact on nostalgia emotion ($\beta = 0.639$, P<0.01). Table 5 also shows that both model 1 and model 2 pass the F-test, and $\Delta R^2$ of model 2 is greater than 0, indicating that the explanatory power of the model is gradually increasing. Therefore, H1 passes the test. This means that enhancing tourism destinations image o will increase tourists' sense of nostalgia emotion.

In order to verify the direct impact of tourism destination image and nostalgia emotion on local attachment, this study constructed Model 3, Model 4, and Model 5 to test. Firstly, the

**Table 4. Correlation coefficient and square root of AVE.**

| Variables | 1 | 2 | 3 | 4 | 5 |
|---|---|---|---|---|---|
| 1. Tourism destination image | 0.760 | | | | |
| 2. Nostalgia emotion | 0.511 | 0.752 | | | |
| 3. Local attachment | 0.394 | 0.636 | 0.791 | | |
| 4. Perceived risk | -0.136 | -0.299 | -0.326 | 0.727 | |
| 5. Revisit intention | 0.334 | 0.463 | 0.475 | -0.587 | 0.814 |

Data source: Statistics based on the questionnaire survey data of this study

**Table 5. Model fitting index and adaptation standard.**

| Fitting index | χ2 | df | χ2 /df | P | NNFI | CFI | NFI | GFI | AGFI | RMSEA |
|---|---|---|---|---|---|---|---|---|---|---|
| Index value | 171.039 | 104 | 1.645 | 0.021 | 0.961 | 0.966 | 0.952 | 0.947 | 0.936 | 0.034 |
| Criteria for good model adaptation | No specific require-ments | No specific require-ments | Less than 2 | Less than 0.05 | More than 0.90 | More than 0.90 | More than 0.90 | More than 0.90 | More than 0.90 | Less than 0.05 |

Data source: Statistics based on the questionnaire survey data of this study

control variables were included in Model 3, which showed a weak positive impact of age on local attachment, while other control variables had no statistically significant impact on local attachment. Secondly, this study incorporated tourism destination image into Model 4 using a hierarchical approach, and the results showed that tourism destination image had a significant positive impact on local attachment (β = 0.575, P<0.05), H2 passes the test. This study incorporated nostalgia emotion into Model 5, and the results showed that nostalgia emotion had a significant positive impact on local attachment (β = 0.713, P<0.001), H3 passed the test. According to the validation results of this research hypothesis, in tourism consumption, when the nostalgia emotion of tourists increases, it will increase their attachment to the place. This is

**Table 6. Multilevel regression analysis.**

| Variables | Nostalgia emotion | | | Local attachment | | | Revisit Intention | |
|---|---|---|---|---|---|---|---|---|
| | Model 1 | Model 2 | Model 3 | Model 4 | Model 5 | Model 6 | Model 7 | Model 8 |
| Intercept | 3.392** (0.023) | 2.736* (0.049) | 3.867* (0.061) | 2.283* (0.093) | 3.029* (0.078) | 3.426* (0.111) | 3.404** (0.066) | 3.734**(0.103) |
| **Control variables** | | | | | | | | |
| Gender | 0.071 | -0.004 | 0.087 | 0.022 | 0.029 | 0.187 | 0.003 | 0.054 |
| From region | 0.142 | 0.019 | 0.019 | 0.035 | 0.022 | 0.137 | 0.027 | 0.022 |
| Age | 0.004* | 0.049 | 0.083* | -0.001 | 0.091 | 0.019 | 0.095 | -0.063 |
| Education | 0.127 | 0.044 | -0.012 | 0.088* | 0.025 | 0.205 | 0.034 | 0.014* |
| Single visit time | 0.116* | 0.016 | 0.018 | 0.773 | 0.041 | 0.075 | 0.117 | 0.131* |
| Number of visits | 0.108 | 0.004 | 0.907 | 0.065 | 0.061 | 0.074 | 0.134 | 0.128** |
| **Independent variable** | | | | | | | | |
| Destination image | | 0.639** | | 0.575* | | | | |
| Nostalgia emotion | | | | | 0.713*** | | | |
| Local attachment | | | | | | | | 0.638* |
| **Moderating variables** | | | | | | | | |
| Perceived risk | | | | | | | 0.428* | |
| **Interaction item** | | | | | | | | |
| Local attachment ×Perceived risk | | | | | | | 0.205* | |
| **R²** | 0.035 | 0.232 | 0.193 | 0.134 | 0.301 | 0.311 | 0.149 | 0.232 |
| **ΔR²** | | 0.077 | | 0.022 | 0.215 | | 0.017 | 0.005 |
| **F** | 2.213* | 2.717** | 3.069** | 2.081* | 3.019* | 3.006** | 2.753* | 2.942* |

Note

*p < 0.05

**p < 0.01

***p< 0.001.

Data source: Statistics based on the questionnaire survey data of this study

very common in tourism consumption, for example, if an elderly person misses a tourist destination they visited when they were young, it will increase their attachment to the destination and lead them to take further tourism actions.

In order to examine the impact of local attachment on revisit intention, this study constructed models 6, 7, and 8. Firstly, the control variables were included in Model 6, indicating that the impact of all control variables on revisit intention was not statistically significant. Subsequently, local attachment was included in Model 8, and research confirmed that local attachment had a significant positive impact on revisit intention ($\beta = 0.638$, $P<0.05$), H4 passed the test. This research result indicates that tourists' attachment to a tourist destination will increase their willingness to revisit. It is obvious that local attachment is an important antecedent for tourists to have a desire to revisit. If a tourist has little attachment to a certain tourist destination, their willingness to revisit is very low.

**4.3.2. Moderating effect test.** The moderating effect of this study was also tested through multi-level regression analysis, as shown in Table 6. In Model 7, this study examined the moderating effect of perceived risk between local attachment and revisit intention. From the research results in Table 5, it can be seen that the interaction coefficient between local attachment and perceived risk is negative and statistically significant ($\beta = - 0.205$, $P<0.05$), indicating that perceived risk plays a negative moderating role in the impact of local attachment on revisiting intention, and the research hypothesis H7 is supported.

In order to more clearly demonstrate the moderating effect of perceived risk between local attachment and revisiting intention, this study drew a moderating effect graph, as shown in Fig 2. When perceived risk is low, local attachment has a strong positive impact on revisiting intention ($\beta = 0.502$, $P<0.01$). When the perceived risk is high, local attachment has a moderate negative impact on the revisiting intention ($\beta = - 0.239$, $P<0.05$). Therefore, perceived risk

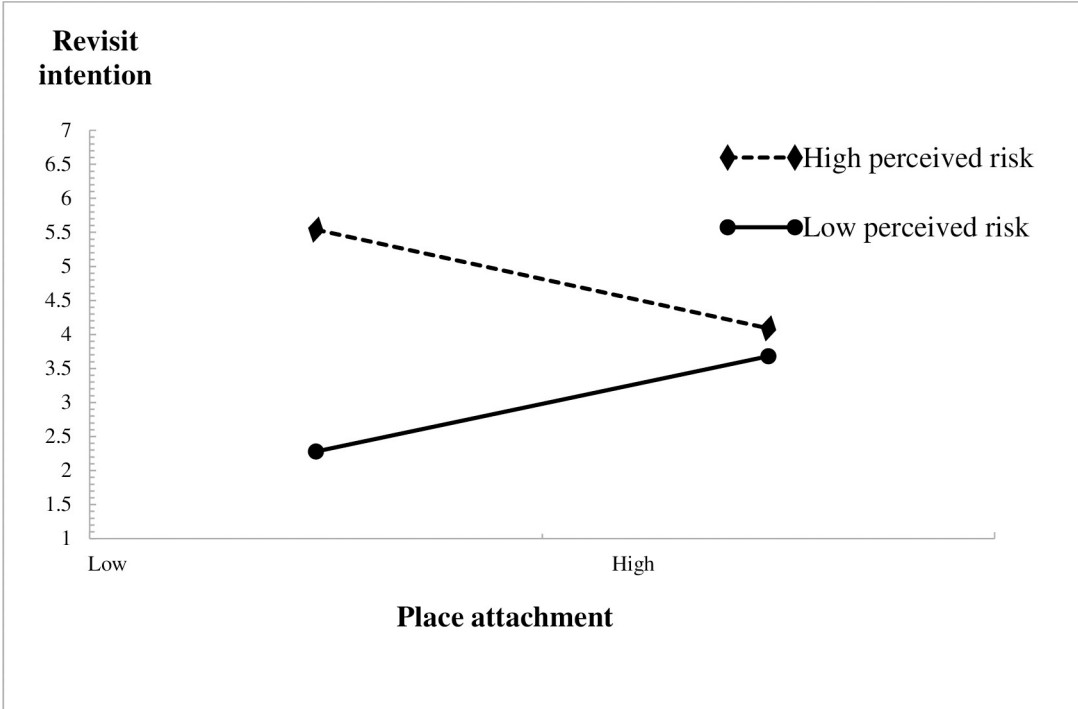

**Fig 2. Moderating effect diagram.**

**Table 7. Comparison of fitness between competition model and chain mediating effect model.**

| Model | χ2 | df | χ2 /df | CFI | TLI | SRMR | RMSEA |
|---|---|---|---|---|---|---|---|
| Chain mediating model | 78.604 | 25 | 3.144 | 0.940 | 0.927 | 0.036 | 0.059 |
| Competition model | 169.187 | 24 | 7.049 | 0.693 | 0.812 | 0.068 | 0.127 |

Data source: Statistics based on the questionnaire survey data of this study

reduces the strong positive impact of local attachment on revisit intention, while perceived risk plays a negative moderating effect. The research hypothesis H7 has been further validated.

This research result indicates that although local attachment has a positive impact on revisit intention, if tourists have perceived risk, it will reduce their revisit intention. For example, when a tourist becomes attached to the Lijiang Scenic Area in Guilin after many years of visiting, it will increase their willingness to revisit. However, if they believe that there is financial or time risk in visiting the scenic area, it will reduce their revisit intention.

**4.3.3. Mediating effect test.**   Preacher et al. believed that the Bootstrap method has obvious advantages in the simple mediating effect test [67]. The two mediating effect hypotheses (H5 and H6) of this study belong to simple mediating effect, so the Bootstrap method will be used to test the mediating effect.

Firstly, in order to further confirm the existence of the chain mediating effect, this study constructs the competition model and the chain mediating effect model, and makes a comparative analysis of the two models. Table 7 shows the fitness indicators of the competition model and the chain mediating effect model. Wu believes that the standard for a good model is χ2/df should be less than 5, CFI and TLI should be greater than 0.90, SRMR should be less than 0.05, and RMSEA should be less than 0.1 [66]. In Table 7, the fitness indicators of the competition model did not meet the good model standards, while the fitness indicators of the chain mediating model met the requirements of the good model, which was significantly better than the competition model. Therefore, this study confirmed the existence of chain mediating effect.

Secondly, this study tested the mediating effect of the two mediation paths. As shown in Table 8, the indirect effect size value of the mediating path "tourism destination image → local attachment → revisit intention" is 0.212, accounting for 29.15% of the total effect, indicating that local attachment plays a mediating effect in the impact of tourism destination image on revisit intention, and the research hypothesis H5 passes the test. The indirect effect size value of the mediating path "tourism destination image → nostalgia emotion → local attachment → revisit intention" is 0.241, accounting for 34.78% of the total effect, indicating that nostalgia emotion and local attachment play a mediating effect in the impact of destination image on revisit intention. The research hypothesis H6 passes the test. As shown in Table 8, the

**Table 8. Mediating effect analysis.**

| Mediating effect | Indirect Effect value | Standard error | Lower limit | Upper limit | Effect proportion |
|---|---|---|---|---|---|
| 1. Tourism destination image → local attachment → revisit intention | 0.212 | 0.003 | 0.101 | 0.397 | 29.15% |
| 2. Tourism destination image → nostalgia emotion → local attachment → revisit intention | 0.241 | 0.027 | 0.123 | 0.488 | 34.78% |
| 3. Total mediating effect | 0.453 | 0.021 | 0.224 | 0.689 | 63.93% |
| 4. Total mediating effect | 0.693 | 0.024 | 0.334 | 0.815 | 100% |

Data source: Statistics based on the questionnaire survey data of this study

confidence intervals of the total mediating effect and the two mediating effects do not contain 0, which are statistically significant. In addition, the total mediating effect of this study is 0.453, accounting for 63.93% of the total effect. The above mediation effect test results indicate that the influence of tourism destination image on revisit intention requires the mediation effect of nostalgia emotion and local attachment, which is a process of emotional transformation of tourists and a mechanism of consumer psychology influence.

## 5. Conclusion and discussion

### 5.1. Research conclusion and theoretical contribution

In order to better compare and analyze the research conclusions with previous related studies, this section discusses the research conclusions and theoretical contributions together. This section will be divided into three sub-sections for discussion. In each sub- section, researchers will summarize the research conclusions and refine the theoretical contributions.

Firstly, tourism destination image has a positive impact on nostalgia emotion and local attachment, and nostalgia emotion has a positive impact on local attachment. H1, H2, and H3 have passed the research hypothesis test. In tourism services, tourism destination image as a result of the interaction between tourists and the destination, is the overall perception of the destination. If the level of tourism destination image is high, it indicates that they have a higher liking for the destination and often reminisce and miss it in the future, making it easier to develop nostalgia emotion and local attachment. In addition, the increase in nostalgia emotion among tourists can make them miss the tourism destination, thereby enhancing the level of local attachment. This study has made certain theoretical contributions to the relationship between tourism destination imager, nostalgia emotion and local attachment. Although previous studies have explored the impact of the above related variables [2, 20], they have all focused on normalized tourism scenarios. However, this study focuses on the tourism scenarios in China after three years of epidemic prevention and control. After China lifted its epidemic prevention and control measures, many tourists have not traveled for a long time, and their psychology has undergone many changes. For example, some tourists are not accustomed to communicating with others and are afraid of socializing. The psychology of tourists is a "black hole", and there are still many unknowns that need to be explored by the theoretical community in order to provide scientific explanations for the practical problems of tourism. Therefore, this study focuses on consumer psychology issues in tourism management and has obtained interdisciplinary research results in management and psychology, which is conducive to enriching the theory of tourism consumer psychology.

Secondly, this study confirms that tourists' local attachment has a positive impact on revisit intention, while perceived risk plays a negative moderating effect the impact of local attachment on revisit intention. H4 and H7 have passed the research hypothesis test. This research result indicates that if tourists develop attachment to a certain tourism destination, their revisit intention will increase. However, when tourists believe that there are certain risks to the destination, such as safety risks, epidemic risks, financial risks, etc., they will reduce revisit intention. In previous studies, it has been confirmed that destination trust, tourism motivation, and communication intention will affect revisit intention [40, 41], and the perceived risk of potential tourists has a significant negative impact on behavioral intention [53]. However, there is still limited literature in tourism research that incorporates local attachment, revisit intention, and perceived risk into the same research framework, and explores the negative moderating effect of perceived risk on the impact of local attachment on revisit intention. Therefore, this study achieved innovation in the research framework and content, added new variables, and expanded the research on the antecedents of

revisit intention. This will help to better explain the impact mechanism of revisit intention and enrich tourism consumption theory.

Thirdly, the research results show that local attachment plays a mediating effect in the impact of tourism destination image on revisit intention. In addition, nostalgia emotion and local attachment play a mediating effect in the impact of destination images on the revisit intention, H5 and H6 passed the research hypothesis test. From the above research results, we can see that tourism destination image has an indirect impact on revisit intention, while nostalgia and local attachment play a mediating effect. In a theoretical sense, the above research conclusions are the development of the research conclusions of Aghnia & Ratiwi, Swets & Cox, Luo [4, 19, 32], because their research only discusses the relationship between certain two variables, and does not put several variables into a research frame to discuss the mediating effect, so their research vision is not wide enough. Therefore, there is relatively little research on the mediating effects of relevant variables based on previous literature, while this study conducted a more in-depth examination and analysis of the mediating effects of nostalgia emotion and local attachment on the impact of tourism destination image on revisit intention. This will help to explain the relationship between the four variables more clearly, facilitate a clearer portrayal of the psychological cognition of tourism consumers, and better understand the formation mechanism of revisit intention.

## 5.2. Practical implications

Firstly, tourist destinations should combine their own reality and find ways to improve their image. The government's tourism management department should introduce some effective manage. Because the factors that affect tourism destination image include natural environment, cultural atmosphere, tourism services, personal experience, etc., the government and tourism enterprises of tourism destinations should improve in these aspects. In terms of natural environment, it is necessary to highlight the local characteristics of the natural environment. For example, the Lijiang Scenic Area in Guilin should highlight its own unique landscape and karst ground, leaving a deep and beautiful impression on tourists. In terms of cultural atmosphere, during the tourism process, tourists should experience rich cultural connotations, historical sites, local characteristics, etc., in order to leave a deep impression on the tourist destination. Improving the quality of tourism services is an important measure to enhance t tourism destination image. It is necessary to enhance employees' service awareness and capabilities through training, and launch distinctive and personalized tourism services. In terms of personal experience, tourists should have a unique experience during the tourism process, participate in local cultural activities (such as the "Zhuang ethnic folk song" in the Lijiang Scenic Area of Guilin), experience local cuisine and clothing, etc., so as to leave a deep impression on the tourist destination and improve the image level of the tourist destination. Through the above measures, we aim to enhance the tourism destination image and create favorable conditions for enhancing tourists' revisit intention. The tourism industry is the pillar industry of Guilin City. The government's tourism management department should start from the long-term tourism development of the city, scientifically plan the development goals of each tourist destination (tourist area), and strengthen strategic management. The tourism management department should vigorously strengthen service quality management and brand management, and introduce a series of regulations on service quality management and brand management. Guilin is a city with rich historical and cultural resources. Tourism management should strengthen the education and promotion of Guilin's history and culture among all tourism practitioners in the city, and improve their cultural literacy. Tourism management also

needs to learn macro management methods from famous foreign tourist cities to improve the effectiveness and scientificity of tourism macro management.

Secondly, it is necessary to guide tourists' emotions and enhance their nostalgia emotion and local attachment. The empirical analysis of this study shows that tourism destination imager has a positive impact on both nostalgia emotion and local attachment, and nostalgia emotion has a positive impact on local attachment. Therefore, tourism destination image plays an important role in enhancing nostalgia emotion and local attachment. As analyzed earlier, it is necessary to enhance tourism destination imager from aspects such as natural environment, cultural atmosphere, tourism services, and personal experience. During the tourism process, tourism companies enhance tourists' nostalgia emotion and local attachment by giving them souvenirs and actively inviting them to participate in tourism services. After tourists end their travel, the government departments and tourism enterprises of the tourism destination can promote the latest tourism services and preferential measures of the tourist destination to tourists through social media such as Weibo, Tiktok and WeChat, so as to stimulate tourists' nostalgia emotion and local attachment.

Thirdly, it is necessary to reduce tourists' perceived risks and enhance their revisit intention. After the control of the COVID-19 was liberalized, China's tourism industry has recovered rapidly, but have some chaos, such as price hikes, poor service quality, and forced tourists to shop. If tourists learn about the chaos of the tourist destination through the media, they will believe that there is uncertainty in the destination, which will have a negative impact on tourists' emotions and create perceived risks. To reduce the perceived risk of tourists, first of all, the government departments of tourist destinations should regulate the tourism order of tourism destinations. Tourism enterprises in tourism destinations should increase price transparency, not arbitrarily increase prices, and eliminate illegal behaviors such as forced shopping. Hotels and tourism enterprises that violate regulations should be fined or closed for rectification. Secondly, tourism enterprises in tourism destinations should improve the quality of tourism services, operate honestly and in a standardized manner, and avoid and reduce tourism service failures. If there is a failure in tourism services, timely service recovery should be carried out, and material and spiritual compensation should be given to tourists to prevent the negative spillover effect of service failure.

## 5.3. Research limitations and prospects

Firstly, in terms of research content, this study explores the impact of tourism destination image, nostalgia emotion and local attachment on intention based on China's tourism consumption scenarios, and tests the moderating effect of perceived risk. However, in the practice of tourism services, tourists' emotions, tourists' sense of face, tourists' emotional intelligence, and employees' Emotional labor will all affect their willingness to revisit. Therefore, in future research, we will attempt to incorporate the above variables for empirical research, and conduct in-depth research on the impact mechanism of revisit intention in tourist destinations.

Secondly, in terms of research methods. This study used a questionnaire survey method to collect research data. Due to the questionnaire designer's pre-set range of questions, the respondents' responses were limited to some extent. In addition, the questionnaire responses will be influenced by the participants' cultural background, region of origin, educational background, mood, cognition, and experience. Future research can use methods such as in-depth interviews and employee log sampling to collect more detailed and in-depth information, improving the reliability of the research.

Thirdly, in terms of sample sources. Because the researchers come from universities in China, it is more convenient to obtain research samples in China. Therefore, the survey

samples for this study are sourced from Chinese citizens and farmers, but there is a lack of samples from countries such as the Americas, Europe, and Southeast Asia, which will affect the generalizability and adaptability of the research conclusions. In future research, we will collaborate with international academic colleagues to expand sample sources and strive to improve the shortcomings of unreasonable sample structures, making the research conclusions more generalizable and adaptable.

## Supporting information

**S1 Appendix.**
(DOCX)

**S1 Data.**
(XLSX)

## Author Contributions

**Conceptualization:** Jiahua Wei, Lei Li.

**Data curation:** Jiahua Wei, Lei Li.

**Formal analysis:** Jiahua Wei.

**Funding acquisition:** Jiahua Wei.

**Investigation:** Jiahua Wei, Lewei Zhou.

**Methodology:** Jiahua Wei, Lewei Zhou, Lei Li.

**Project administration:** Jiahua Wei, Lei Li.

**Resources:** Jiahua Wei.

**Software:** Jiahua Wei, Lewei Zhou.

**Supervision:** Jiahua Wei.

**Validation:** Jiahua Wei.

**Visualization:** Jiahua Wei.

**Writing – original draft:** Jiahua Wei.

**Writing – review & editing:** Jiahua Wei, Lei Li.

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
