## [Decision Letter · Decision Letter 0]

13 Nov 2023

PONE-D-23-31058A Study on the Impact of Tourism Destination Image and Local Attachment on the Revisit Intention: The Moderating Effect of Perceived RiskPLOS ONE

Dear Dr. Wei,

Thank you for submitting your manuscript to PLOS ONE. After careful consideration, we feel that it has merit but does not fully meet PLOS ONE’s publication criteria as it currently stands. Therefore, we invite you to submit a revised version of the manuscript that addresses the points raised during the review process.

We look forward to receiving your revised manuscript.

Kind regards,

Bo Pu, Ph.D.

Academic Editor

PLOS ONE

Journal Requirements:

"1. Guangxi Natural Science Foundation（Grant No. 2021GXNSFAA220066).

2. National Natural Science Foundation of China (Grant No. 72262009)."

Additional Editor Comments:

Please revise this manuscript according to the comments from the reviews.

Reviewers' comments:

Reviewer's Responses to Questions

**Comments to the Author**

1. Is the manuscript technically sound, and do the data support the conclusions?

Reviewer #1: Yes

Reviewer #2: No

Reviewer #3: Yes

Reviewer #4: Yes

Reviewer #5: Yes

2. Has the statistical analysis been performed appropriately and rigorously? 

Reviewer #1: Yes

Reviewer #2: No

Reviewer #3: Yes

Reviewer #4: I Don't Know

Reviewer #5: Yes

3. Have the authors made all data underlying the findings in their manuscript fully available?

Reviewer #1: Yes

Reviewer #2: No

Reviewer #3: Yes

Reviewer #4: No

Reviewer #5: Yes

4. Is the manuscript presented in an intelligible fashion and written in standard English?

Reviewer #1: Yes

Reviewer #2: No

Reviewer #3: Yes

Reviewer #4: Yes

Reviewer #5: Yes

5. Review Comments to the Author

Reviewer #1: The importance of a manuscript for the scientific community depends on various factors such as the novelty of the research, the significance of the findings, and the potential impact on the field. the manuscript addresses a gap in knowledge, presents groundbreaking findings, and offers new insights that can expand existing knowledge.

Reviewer #2: In review of this manuscript, several concerns have been identified, detailed as follows:

1. The abstract does not adequately engage with existing literature to highlight the novelty of the manuscript.

2. The literature review section appears to be a mere enumeration of existing works, failing to underscore the innovative aspects of the manuscript.

3. The research design section lacks rigor. The questionnaire data presented do not sufficiently establish causality to yield empirical findings.

4. The data analysis section would benefit from a more comprehensive discussion.

5. The conclusion section appears to overstate its novelty, particularly in terms of the research scenarios. Additionally, the manuscript would benefit from refinements in language and formatting.

Reviewer #3: This study takes China’s ecological tourism scenic area (Guilin Lijiang Scenic Area) as a case study, and uses questionnaire survey method to obtain research data for empirical research. Well-written and structured paper. Methodology is correct. Results are useful and novelty. The results of this study confirm that tourism destination image has a positive impact on nostalgia emotions and local attachment, nostalgia emotion has a positive impact on local attachment, and local attachment has a positive impact on revisit intention. The discussion and conclusion stated that tourism destination image has a positive impact on nostalgia emotion and local attachment, and nostalgia emotion has a positive impact on local attachment. Moreover, this study confirms that tourists’ local attachment has a positive impact on revisit intention, while perceived risk plays a negative moderating effect the impact of local attachment on revisit intention. Finally, the research results show that local attachment plays a mediating effect in the impact of tourism destination image on revisit intention.

Recommendations:

1. Literature review and research hypotheses have to be separated.

2. The sources of tables and figures are missing (E.g. own compilation, etc.).

3. The Literature review has to be extended; more citations are needed.

E.g.

https://www.tandfonline.com/doi/full/10.1080/23311886.2023.2240569

and

https://www.mdpi.com/2071-1050/13/12/6704

and

https://www.mdpi.com/2071-1050/14/17/10716

Etc.

Reviewer #4: This manuscript presents a complex model of tourist revisiting intentions, using multiple predictors and indirect paths. It is a worthy attempt in defining the determinants of tourist loyalty, but there are many issues to be addressed, before it can be accepted for publication:

1. Literature, from which the authors are drawing their hypotheses, is mostly from Chinese and Far East authors, and seems to be somewhat incidentally collected - based on what was easily available to the authors, at the time of writing the manuscript. It would be expected to see at least some studies from the marketing field, on satisfaction and loyalty, since revisiting a tourist destination is a classic measure of attitudinal tourist loyalty.

2. The model building shows a high level of ambition, as multiple indirect paths were included, with both moderation and mediation, which is good.

3. The research sample is quite restricted, in terms of generalizing the results. Authors discuss this correctly as a limitation of their research, but it should be also shown how the cultural factors might influence the results. Namely, the questionnaire, obviously translated from Chinese to English, might have a completely different meaning and cultural context implied in the Chinese original. e.g.: "Guilin Lijiang Scenic Area makes me Relax and Relax" (I wonder what this could mean in English, compared to Chinese?) and "My heart still stays in Guilin

Lijiang Scenic Area".

4. The questionnaire consists of multiple items, and we have no idea what is their source. Have the been used before in verified research instruments? Authors provide an adequate analysis of reliability and validity, but they should still declare how the questionnaire items were developed.

5. Presentation of results and their statistical analysis is quite unorthodox. This reviewer is not an expert in quantitative analysis, but it is very difficult to acknowledge a mix of regression-based techniques and Structural Equation Modelling (SEM), used for analysis of different hypotheses subsets. It would be much easier to use a single method of analysis, such as regression-based Hayes PROCESS macro for SPSS, or AMOS for SEM, since authors have declared to use both SPSS and AMOS in their analysis. SEM has not even been mentioned as a method of data analysis. A reviewer more knowledgeable in complex quantitative methods than myself should be consulted to evaluate if the current mix of two methods is consisent and acceptable. However, I can say that the presentation of results is substandard, while the mix of two different methods is not seen in the major research journals.

6. Discussion of results is substandard as well, which flows directly from a not so appropriate selection of literature. In the discussion section, there is another unusual translation from English: "Practical englightment" should be, probably, translated in terms of "Practical implications".

All these issues should be addressed in a major revision of the manuscript.

Reviewer #5: In the manuscript, the effect of tourism destination image on nostalgia emotion and the effect of these two factors on local attachment were investigated. It has been determined that these factors are effective. The moderating effect of perceived risk on the effect of local attachment on revisit intention was examined.

The negative moderating effect of Perceived risk was determined.

The hypotheses of the research and the literature are very well designed. An appropriate method was used in the analysis of the data obtained from the research and was explained in full detail. The importance of the subject of the research was clearly discussed both in terms of literature and field research findings. The clear presentation of all items and the comprehensibility of the tables are positively noteworthy.

In the discussion part of the research, the contribution of the research to the literature and the tourism sector should be emphasized more. The findings of a well-designed study need to be expressed in a more striking way. In particular, research suggestions that guide future researchers should be emphasized more.

I wish you good luck.

6. PLOS authors have the option to publish the peer review history of their article (what does this mean?). If published, this will include your full peer review and any attached files.

Reviewer #1: No

Reviewer #2: No

Reviewer #3: No

Reviewer #4: No

Reviewer #5: No

---

## [Author Response · Author response to Decision Letter 0]

29 Nov 2023

Response to Reviewers

Dear reviewers,

We write in response to the modifications made to the manuscript “A Study on the Impact of Tourism Destination Image and Local Attachment on the Revisit Intention: The Moderating Effect of Perceived Risk” submitted for publication. Based on the reviewer’ recommendations made by the reviewers, the manuscript has been revised taking cognizance of the insightful comments. All revisions (include additional references) in the manuscript are marked with red color. The comments (in red color) by the reviewers and the response (in blue) by authors are found below.

 We would like also to thank you for allowing us to resubmit a revised copy of the manuscript. We hope that the revised manuscript is accepted for publication in the PLOS ONE.

Response to reviewers’ comments

Reviewer: #1: 

The importance of a manuscript for the scientific community depends on various factors such as the novelty of the research, the significance of the findings, and the potential impact on the field. the manuscript addresses a gap in knowledge, presents groundbreaking findings, and offers new insights that can expand existing knowledge.

Reply:

Thank you very much to the reviewing experts for their affirmation and recognition of the paper. This is an encouragement for our future research and has given us the motivation to further explore tourism issues. Since you did not provide specific modification suggestions, we can only make modifications based on the suggestions of the other two experts this time. We look forward to receiving your guidance in future research!

Reviewer #2: 

 In review of this manuscript, several concerns have been identified, detailed as follows:

1. The abstract does not adequately engage with existing literature to highlight the novelty of the manuscript.

Reply: 

Thank you very much for the suggestions from the reviewing experts. In the revised manuscript, we have made revisions to the abstract, highlighting the innovation and research value of the paper. However, according to the writing method of the paper, there is no need to introduce the current literature in the abstract. In the introduction, we specifically analyzed the shortcomings of current literature research and analyzed the innovation of our research. The revised summary is as follows (P1):

“The revisit intention of tourists is an important guarantee for the sustainable and healthy development of tourism destination, and has also received attention from the current academic community. However, there is still insufficient research on the antecedents of revisit intention from the perspectives of tourism destination, image and nostalgia emotion.This study takes China’s ecological tourism scenic area (Guilin Lijiang Scenic Area) as a case study, and uses questionnaire survey method to obtain research data for empirical research. The results of this study confirm that tourism destination image has a positive impact on nostalgia emotions and local attachment, nostalgia emotion has a positive impact on local attachment, and local attachment has a positive impact on revisit intention. Perceived risk plays a negative moderating effect between local attachment and revisit intention. In addition, this study also examined the mediating effect of nostalgia emotion and local attachment. This study is beneficial for enriching the theory of the influence mechanism of revisit intention from the perspective of consumer psychology. It is an interdisciplinary research result of management and psychology, providing theoretical reference for improving revisit intention in tourism destinations and promoting their healthy development.”

2. The literature review section appears to be a mere enumeration of existing works, failing to underscore the innovative aspects of the manuscript.

Reply:

Dear reviewers, regarding the innovation of the paper, we mainly focus on “Introduction” and “5.1. Analyze the research conclusion and theoretical contribution”. In the introduction, we discuss as follows:

“…This study will enrich the research theory of consumer psychology, expand the research perspective of tourism management, provide theoretical reference and practical guidance for tourism destinations and tourism enterprises to improve their brand image and revisit intention, and promote the healthy development of China and even the world’s tourism industry.”

 Based on the suggestions of the reviewing experts and the standardized writing methods of the paper, we analyzed the shortcomings of the current research in the literature review and also mentioned the innovation of our research on a certain variable. Thank you again for the suggestions from the reviewing experts.

3. The research design section lacks rigor. The questionnaire data presented do not sufficiently establish causality to yield empirical findings.

Reply:

Dear reviewer, thank you for your question. I will provide a serious explanation as follows. Our research design referred to commonly used methods in current questionnaire surveys (Preacher et al., 2007; Huang et al., 2016; Wu, 2010;). Our survey sample is 367, and the questionnaire items are 24, The sample is 15.29 times larger than the questionnaire，which meets the requirement of Wu (2010) that the survey sample is more than 10 times the number of questionnaire items. In the descriptive analysis, reliability and validity analysis, and multiple regression analysis of the paper, we followed the standardized data analysis of SPSS and AMOS. Thank you very much for the questions from the reviewing experts

4. The data analysis section would benefit from a more comprehensive discussion.

Reply: 

Thank you very much for the questions from the reviewing experts. For the data analysis of the paper, we have provided a more in-depth discussion and explanation in the revised manuscript. For example, in the reliability and validity analysis section, we discussed and explained as follows:

“…As shown in Table 6, Model 1 is a regression of nostalgia emotion using control variables. The research results show that, except for the weak impact of age and single visit frequency on nostalgia emotion, other control variables have no statistically significant impact on nostalgia emotion. Model 2 shows that tourism destination image has a significant positive impact on nostalgia emotion（β= 0.639, P<0.01). Table 5 also shows that both model 1 and model 2 pass the F-test, and Δ R2 of model 2 is greater than 0, indicating that the explanatory power of the model is gradually increasing. Therefore, H1 passes the test. This means that enhancing tourism destinations image o will increase tourists' sense of nostalgia emotion.

In order to verify the direct impact of tourism destination image and nostalgia emotion on local attachment, this study constructed Model 3, Model 4, and Model 5 to test. Firstly, the control variables were included in Model 3, which showed a weak positive impact of age on local attachment, while other control variables had no statistically significant impact on local attachment. Secondly, this study incorporated tourism destination image into Model 4 using a hierarchical approach, and the results showed that tourism destination image had a significant positive impact on local attachment (β=0.575, P<0.05), H2 passes the test. This study incorporated nostalgia emotion into Model 5, and the results showed that nostalgia emotion had a significant positive impact on local attachment（β= 0.713, P<0.001), H3 passed the test. According to the validation results of this research hypothesis, in tourism consumption, when the nostalgia emotion of tourists increases, it will increase their attachment to the place. This is very common in tourism consumption, for example, if an elderly person misses a tourist destination they visited when they were young, it will increase their attachment to the destination and lead them to take further tourism actions.

In order to examine the impact of local attachment on revisit intention, this study constructed models 6, 7, and 8. Firstly, the control variables were included in Model 6, indicating that the impact of all control variables on revisit intention was not statistically significant. Subsequently, local attachment was included in Model 8, and research confirmed that local attachment had a significant positive impact on revisit intention（β= 0.638, P<0.05), H4 passed the test. This research result indicates that tourists' attachment to a tourist destination will increase their willingness to revisit. It is obvious that local attachment is an important antecedent for tourists to have a desire to revisit. If a tourist has little attachment to a certain tourist destination, their willingness to revisit is very low.”

I also had an in-depth discussion on the analysis results of the direct impact test, moderation effect test, and mediation effect test (P14-P19). In addition, we also discussed the data analysis results in the research conclusion (P20-P22). Thank you again for the suggestions from the reviewing experts.

5. The conclusion section appears to overstate its novelty, particularly in terms of the research scenarios. Additionally, the manuscript would benefit from refinements in language and formatting.

Reply: 

Thank you very much for the suggestions from the reviewers. Based on your suggestions, in the revised manuscript, we have addressed section “5.1 The research conclusion and theoretical contribution” have been revised to objectively evaluate the innovation of the paper（P20-P22）. The revised discussion is as follows：

“Firstly, tourism destination image has a positive impact on nostalgia emotion and local attachment, and nostalgia emotion has a positive impact on local attachment. H1, H2, and H3 have passed the research hypothesis test. In tourism services, tourism destination image as a result of the interaction between tourists and the destination, is the overall perception of the destination. If the level of tourism destination image is high, it indicates that they have a higher liking for the destination and often reminisce and miss it in the future, making it easier to develop nostalgia emotion and local attachment. In addition, the increase in nostalgia emotion among tourists can make them miss the tourism destination, thereby enhancing the level of local attachment. This study has made certain theoretical contributions to the relationship between tourism destination imager, nostalgia emotion and local attachment. Although previous studies have explored the impact of the above related variables (Fan et al., 2014; Yu, 2022), they have all focused on normalized tourism scenarios. However, this study focuses on the tourism scenarios in China after three years of epidemic prevention and control. After China lifted its epidemic prevention and control measures, many tourists have not traveled for a long time, and their psychology has undergone many changes. For example, some tourists are not accustomed to communicating with others and are afraid of socializing. The psychology of tourists is a “black hole”, and there are still many unknowns that need to be explored by the theoretical community in order to provide scientific explanations for the practical problems of tourism. Therefore, this study focuses on consumer psychology issues in tourism management and has obtained interdisciplinary research results in management and psychology, which is conducive to enriching the theory of tourism consumer psychology.

Secondly, this study confirms that tourists’ local attachment has a positive impact on revisit intention, while perceived risk plays a negative moderating effect the impact of local attachment on revisit intention. H4 and H7 have passed the research hypothesis test. This research result indicates that if tourists develop attachment to a certain tourism destination, their revisit intention will increase. However, when tourists believe that there are certain risks to the destination, such as safety risks, epidemic risks, financial risks, etc., they will reduce revisit intention. In previous studies, it has been confirmed that destination trust, tourism motivation, and communication intention will affect revisit intention (Abubakar et al., 2017; Zhao et al., 2019), and the perceived risk of potential tourists has a significant negative impact on behavioral intention (Xie et al., 2020). However, there is still limited literature in tourism research that incorporates local attachment, revisit intention, and perceived risk into the same research framework, and explores the negative moderating effect of perceived risk on the impact of local attachment on revisit intention. Therefore, this study achieved innovation in the research framework and content, added new variables, and expanded the research on the antecedents of revisit intention. This will help to better explain the impact mechanism of revisit intention and enrich tourism consumption theory.

Thirdly, the research results show that local attachment plays a mediating effect in the impact of tourism destination image on revisit intention. In addition, nostalgia emotion and local attachment play a mediating effect in the impact of destination images on the revisit intention, H5 and H6 passed the research hypothesis test. From the above research results, we can see that tourism destination image has an indirect impact on revisit intention, while nostalgia and local attachment play a mediating effect. In a theoretical sense, the above research conclusions are the development of the research conclusions of Aghnia & Pratiwi (2023), Swets & Cox (2022), Luo (2016) and others, because their research only discusses the relationship between certain two variables, and does not put several variables into a research frame to discuss the mediating effect, so their research vision is not wide enough. Therefore, there is relatively little research on the mediating effects of relevant variables based on previous literature, while this study conducted a more in-depth examination and analysis of the mediating effects of nostalgia emotion and local attachment on the impact of tourism destination image on revisit intention. This will help to explain the relationship between the four variables more clearly, facilitate a clearer portrayal of the psychological cognition of tourism consumers, and better understand the formation mechanism of revisit intention.”

Thank you again for the suggestions from the reviewing experts.

Reviewer #3:

 This study takes China’s ecological tourism scenic area (Guilin Lijiang Scenic Area) as a case study, and uses questionnaire survey method to obtain research data for empirical research. Well-written and structured paper. Methodology is correct. Results are useful and novelty. The results of this study confirm that tourism destination image has a positive impact on nostalgia emotions and local attachment, nostalgia emotion has a positive impact on local attachment, and local attachment has a positive impact on revisit intention. The discussion and conclusion stated that tourism destination image has a positive impact on nostalgia emotion and local attachment, and nostalgia emotion has a positive impact on local attachment. Moreover, this study confirms that tourists’ local attachment has a positive impact on revisit intention, while perceived risk plays a negative moderating effect the impact of local attachment on revisit intention. Finally, the research results show that local attachment plays a mediating effect in the impact of tourism destination image on revisit intention.

Recommendations:

1. Literature review and research hypotheses have to be separated.

Reply:

Thank you very much to the reviewers for their evaluation and suggestions on the paper. I referred to the practices of Gavin et al. (2020), Huang et al. (2022), Jung et al. (2017), Zhang and Wang (2022) to write a literature review and research hypotheses together. Because in this study, the literature review and research hypotheses are both regression and analysis of previous literature, and separating them may feel less logical. For example, if a previous literature contains a relationship between two variables, and the literature review and research hypothesis are separated and placed in the literature review section instead of the research hypothesis, the research hypothesis will lack persuasiveness. Many excellent papers also combine literature reviews with research hypotheses. Thank you again for the suggestions from the reviewing experts. It is my pleasure to discuss academic issues.

2. The sources of tables and figures are missing (E.g. own compilation, etc.).

Reply:

I am very grateful for the reviewer's suggestions. In the revised draft, we have annotated the data statistics table as follows:“Data source: Statistics based on the questionnaire survey data of this study.”

3. The Literature review has to be extended; more citations are needed.

E.g.

https://www.tandfonline.com/doi/full/10.1080/23311886.2023.2240569

and

https://www.mdpi.com/2071-1050/13/12/6704

and

https://www.mdpi.com/2071-1050/14/17/10716

Etc.

Reply:

Thank you very much for the expert's advice. I have read the above three references, which are very helpful for my research. I have cited the above three references in the revised manuscript. Thank you very much for the suggestions from the reviewing experts!

Reviewer #4: 

This manuscript presents a complex model of tourist revisiting intentions, using multiple predictors and indirect paths. It is a worthy attempt in defining the determinants of tourist loyalty, but there are many issues to be addressed, before it can be accepted for publication:

1. Literature, from which the authors are drawing their hypotheses, is mostly from Chinese and Far East authors, and seems to be somewhat incidentally collected - based on what was easily available to the authors, at the time of writing the manuscript. It would be expected to see at least some studies from the marketing field, on satisfaction and loyalty, since revisiting a tourist destination is a classic measure of attitudinal tourist loyalty.

Reply: 

Thank you very much for the suggestions from the reviewing experts. Our selection of literature is based on the correlation between the literature and this study. Based on your suggestion, we have also added literature on customer loyalty and sustained trust in the “revisit intention” section. In the first paragraph of this section, we have added the following discussion:

“The revisit intention indicates a loyalty of tourists to the destination (Ranjbarian, 2015).Customer loyalty refers to the preference and repetitive purchasing behavior of customers towards a company’s products or services over a long period of time (Wei and Lin, 2022). The willingness to revisit reflects a loyalty of tourists to the destination (Ranjbarian, 2015). The willingness to revisit is a manifestation of consumers’ sustained trust. Consumer sustained trust refers to the trust generated after the first transaction between consumers and businesses, which is divided into three dimensions: sustained ability, sustained integrity, and sustained goodwill (Huang and Chang, 2020). Shankar et al. (2002) summarized three types of factors that affect online shopping trust, including consumer characteristics, merchant characteristics, and interaction factors”.

2. The model building shows a high level of ambition, as multiple indirect paths were included, with both moderation and mediation, which is good.

Reply:

Thank you very much for the affirmation from the reviewing experts. We have been engaged in research on tourism consumption psychology, mainly obtaining research data through questionnaire surveys and situational experiments. In the past two years, we have published about 10 SSCI and SCI papers. This paper is also one of our series of studies. Our research model is also an extension of our previous research. We have referred to the shortcomings of previous literature research and proposed this research model. Thank you again for the expert’s affirmation, and we look forward to further communication in the future.

3. The research sample is quite restricted, in terms of generalizing the results. Authors discuss this correctly as a limitation of their research, but it should be also shown how the cultural factors might influence the results. Namely, the questionnaire, obviously translated from Chinese to English, might have a completely different meaning and cultural context implied in the Chinese original. e.g.: "Guilin Lijiang Scenic Area makes me Relax and Relax" (I wonder what this could mean in English, compared to Chinese?) and "My heart still stays in Guilin Lijiang Scenic Area".

Reply:

Respected reviewer. We discussed the limitations and future improvement directions of the questionnaire survey in section “5.3. Research Limitations and Prospects”（P23）: 

“this study explores the impact of tourism destination image, nostalgia emotion and local attachment on intention based on China’s tourism consumption scenarios, and tests the moderating effect of perceived risk. However, in the practice of tourism services, tourists’ emotions, tourists’ sense of face, tourists’ emotional intelligence, and employees’ Emotional labor will all affect their willingness to revisit. Therefore, in future research, we will attempt to incorporate the above variables for empirical research, and conduct in-depth research on the impact mechanism of revisit intention in tourist destinations.”

But in terms of research samples, our sample is sufficient. According to Wu (2010) and Nunnally (1994), in a questionnaire survey, when the sample size is more than 10 times the number of questionnaire items, the sample is considered sufficient. In our study, there were a total of 24 items in the questionnaire, and the sample size was 367, which is 15.29 times larger than the questionnaire items.

For the expression of questionnaire items, as our questionnaire items are intended for Chinese tourists, our expression should be close to the cultural background of China. For translation, we also sought advice from English tour guides in Guilin tourist attractions and polished it with a PhD in English. Thank you again for your suggestion.

4. The questionnaire consists of multiple items, and we have no idea what is their source. Have the been used before in verified research instruments? Authors provide an adequate analysis of reliability and validity, but they should still declare how the questionnaire items were developed.

Reply:

Dear reviewer, when designing the survey questionnaire, we referred to mature scales from relevant literature for all 5 variables. The literature sources for the questionnaire are shown in Table 1 of the paper (P8-P9). We have discussed the following in our paper:

“…The measurement items of the five variables in this study are all sourced from authoritative measurement scales currently available. On this basis, researchers made necessary adjustments and modifications to the textual narrative style and some items of the scale through expert interviews and small sample testing based on the specific scenario of this study, thus forming a formal measurement scale. The sources of measurement items for the 5 variables are shown in Table 1.

All measurement items were measured using the Likert 5-point scale. During the questionnaire development process, researchers converted the items in the English scale into Chinese and then translated them into English through a standard translation back translation process. In order to further ensure the effectiveness of the questionnaire, researchers specifically consulted scholars and industry experts in the fields of tourism management and psychology on whether the questionnaire items were reasonable before conducting formal research. Based on this, the questionnaire items were adjusted and formed a formal questionnaire. The survey questionnaire also set measurement items such as gender, age, source, education, occupation, etc. to investigate the personal situation of the participants. Please refer to the appendix for the questionnaire.”

5. Presentation of results and their statistical analysis is quite unorthodox. This reviewer is not an expert in quantitative analysis, but it is very difficult to acknowledge a mix of regression-based techniques and Structural Equation Modelling (SEM), used for analysis of different hypotheses subsets. It would be much easier to use a single method of analysis, such as regression-based Hayes PROCESS macro for SPSS, or AMOS for SEM, since authors have declared to use both SPSS and AMOS in their analysis. SEM has not even been mentioned as a method of data analysis. A reviewer more knowledgeable in complex quantitative methods than myself should be consulted to evaluate if the current mix of two methods is consisent and acceptable. However, I can say that the presentation of results is substandard, while the mix of two different methods is not seen in the major research journals.

Reply:

Dear reviewer, many survey data analyses are now conducted in conjunction with SPSS and AMOS. We also referred to the data analysis methods of Wu (2010), Preacher et al. (2007), and others. We think these methods are relatively scientific and reasonable. In our study, we conducted reliability and validity tests on the survey questionnaire data using SPSS 26.0 and AMOS 26.0. Wu (2010) pointed out that using AMOS to test the fit (construct validity) of the model is more suitable, so we used AMOS to test the fit of the model, as shown in Table 5 of the paper (P14). SPSS is not suitable for testing the fit of the model.

In other parts of the paper, a certain method was used for analysis, such as using SPSS software for multi-level regression analysis to test the direct impact relationship and moderating effect. Thank you again for the expert's advice.

6. Discussion of results is substandard as well, which flows directly from a not so appropriate selection of literature. In the discussion section, there is another unusual translation from English: "Practical englightment" should be, probably, translated in terms of "Practical implications".

All these issues should be addressed in a major revision of the manuscript.

Reply:

Thank you very much for the suggestions from the reviewing experts. Firstly, in the revised draft, we have changed "5.2 Practical englight " to "5.2 Practical implications". Secondly, we have made content modifications to "5.2 Practical implications" and delved deeper into the theoretical significance of the paper (P21-P22). For example, in the first paragraph of our section, we have made modifications and discussed the following:

“Firstly, tourist destinations should combine their own reality and find ways to improve their image. The government’s tourism management department should introduce some effective manage. Because the factors that affect tourism destination image include natural environment, cultural atmosphere, tourism services, personal experience, etc., the government and tourism enterprises of tourism destinations should improve in these aspects. In terms of natural environment, it is necessary to highlight the local characteristics of the natural environment. For example, the Lijiang Scenic Area in Guilin should highlight its own unique landscape and karst ground, leaving a deep and beautiful impression on tourists. In terms of cultural atmosphere, during the tourism process, tourists should experience rich cultural connotations, historical sites, local characteristics, etc., in order to leave a deep impression on the tourist destination. Improving the quality of tourism services is an important measure to enhance t tourism destination image. It is necessary to enhance employees’ service awareness and capabilities through training, and launch distinctive and personalized tourism services. In terms of personal experience, tourists should have a unique experience during the tourism process, participate in local cultural activities (such as the “Zhuang ethnic folk song” in the Lijiang Scenic Area of Guilin), experience local cuisine and clothing, etc., so as to leave a deep impression on the tourist destination and improve the image level of the tourist destination. Through the above measures, we aim to enhance the tourism destination image and create favorable conditions for enhancing tourists’ revisit intention. The tourism industry is the pillar industry of Guilin City. The government’s tourism management department should start from the long-term tourism development of the city, scientifically plan the development goals of each tourist destination (tourist area), and strengthen strategic management. The tourism management department should vigorously strengthen service quality management and brand management, and introduce a series of regulations on service quality management and brand management. Guilin is a city with rich historical and cultural resources. Tourism management should strengthen the education and promotion of Guilin's history and culture among all tourism practitioners in the city, and improve their cultural literacy. Tourism management also needs to learn macro management methods from famous foreign tourist cities to improve the effectiveness and scientificity of tourism macro management.”

Reviewer #5:

 In the manuscript, the effect of tourism destination image on nostalgia emotion and the effect of these two factors on local attachment were investigated. It has been determined that these factors are effective. The moderating effect of perceived risk on the effect of local attachment on revisit intention was examined.

The negative moderating effect of Perceived risk was determined.

The hypotheses of the research and the literature are very well designed. An appropriate method was used in the analysis of the data obtained from the research and was explained in full detail. The importance of the subject of the research was clearly discussed both in terms of literature and field research findings. The clear presentation of all items and the comprehensibility of the tables are positively noteworthy.

In the discussion part of the research, the contribution of the research to the literature and the tourism sector should be emphasized more. The findings of a well-designed study need to be expressed in a more striking way. In particular, research suggestions that guide future researchers should be emphasized more.

I wish you good luck.

Reply: 

Thank you to the reviewing experts for their affirmation of the research content and methods of the paper. We have also followed the recommendations of the reviewing experts, in section 5.1, We have made the following modifications (P19-P21): 

“Firstly, tourism destination image has a positive impact on nostalgia emotion and local attachment, and nostalgia emotion has a positive impact on local attachment. H1, H2, and H3 have passed the research hypothesis test. In tourism services, tourism destination image as a result of the interaction between tourists and the destination, is the overall perception of the destination. If the level of tourism destination image is high, it indicates that they have a higher liking for the destination and often reminisce and miss it in the future, making it easier to develop nostalgia emotion and local attachment. In addition, the increase in nostalgia emotion among tourists can make them miss the tourism destination, thereby enhancing the level of local attachment. This study has made certain theoretical contributions to the relationship between tourism destination imager, nostalgia emotion and local attachment. Although previous studies have explored the impact of the above related variables (Fan et al., 2014; Yu, 2022), they have all focused on normalized tourism scenarios. However, this study focuses on the tourism scenarios in China after three years of epidemic prevention and control. After China lifted its epidemic prevention and control measures, many tourists have not traveled for a long time, and their psychology has undergone many changes. For example, some tourists are not accustomed to communicating with others and are afraid of socializing. The psychology of tourists is a “black hole”, and there are still many unknowns that need to be explored by the theoretical community in order to provide scientific explanations for the practical problems of tourism. Therefore, this study focuses on consumer psychology issues in tourism management and has obtained interdisciplinary research results in management and psychology, which is conducive to enriching the theory of tourism consumer psychology.

Secondly, this study confirms that tourists’ local attachment has a positive impact on revisit intention, while perceived risk plays a negative moderating effect the impact of local attachment on revisit intention. H4 and H7 have passed the research hypothesis test. This research result indicates that if tourists develop attachment to a certain tourism destination, their revisit intention will increase. However, when tourists believe that there are certain risks to the destination, such as safety risks, epidemic risks, financial risks, etc., they will reduce revisit intention. In previous studies, it has been confirmed that destination trust, tourism motivation, and communication intention will affect revisit intention (Abubakar et al., 2017; Zhao et al., 2019), and the perceived risk of potential tourists has a significant negative impact on behavioral intention (Xie et al., 2020). However, there is still limited literature in tourism research that incorporates local attachment, revisit intention, and perceived risk into the same research framework, and explores the negative moderating effect of perceived risk on the impact of local attachment on revisit intention. Therefore, this study achieved innovation in the research framework and content, added new variables, and expanded the research on the antecedents of revisit intention. This will help to better explain the impact mechanism of revisit intention and enrich tourism consumption theory.

Thirdly, the research results show that local attachment plays a mediating effect in the impact of tourism destination image on revisit intention. In addition, nostalgia emotion and local attachment play a mediating effect in the impact of destination images on the revisit intention, H5 and H6 passed the research hypothesis test. From the above research results, we can see that tourism destination image has an indirect impact on revisit intention, while nostalgia and local attachment play a mediating effect. In a theoretical sense, the above research conclusions are the development of the research conclusions of Aghnia & Pratiwi (2023), Swets & Cox (2022), Luo (2016) and others, because their research only discusses the relationship between certain two variables, and does not put several variables into a research frame to discuss the mediating effect, so their research vision is not wide enough. Therefore, there is relatively little research on the mediating effects of relevant variables based on previous literature, while this study conducted a more in-depth examination and analysis of the mediating effects of nostalgia emotion and local attachment on the impact of tourism destination image on revisit intention. This will help to explain the relationship between the four variables more clearly, facilitate a clearer portrayal of the psychological cognition of tourism consumers, and better understand the formation mechanism of revisit intention.”

In the first paragraph of "5.2 Practical implications", We made modifications to the content and analyzed the tourism objectives and how the government tourism department should take action (P21-P23). we discussed the following: 

 “Firstly, tourist destinations should combine their own reality and find ways to improve their image. The government’s tourism management department should introduce some effective manage. Because the factors that affect tourism destination image include natural environment, cultural atmosphere, tourism services, personal experience, etc., the government and tourism enterprises of tourism destinations should improve in these aspects. In terms of natural environment, it is necessary to highlight the local characteristics of the natural environment. For example, the Lijiang Scenic Area in Guilin should highlight its own unique landscape and karst ground, leaving a deep and beautiful impression on tourists. In terms of cultural atmosphere, during the tourism process, tourists should experience rich cultural connotations, historical sites, local characteristics, etc., in order to leave a deep impression on the tourist destination. Improving the quality of tourism services is an important measure to enhance t tourism destination image. It is necessary to enhance employees’ service awareness and capabilities through training, and launch distinctive and personalized tourism services. In terms of personal experience, tourists should have a unique experience during the tourism process, participate in local cultural activities (such as the “Zhuang ethnic folk song” in the Lijiang Scenic Area of Guilin), experience local cuisine and clothing, etc., so as to leave a deep impression on the tourist destination and improve the image level of the tourist destination. Through the above measures, we aim to enhance the tourism destination image and create favorable conditions for enhancing tourists’ revisit intention. The tourism industry is the pillar industry of Guilin City. The government’s tourism management department should start from the long-term tourism development of the city, scientifically plan the development goals of each tourist destination (tourist area), and strengthen strategic management. The tourism management department should vigorously strengthen service quality management and brand management, and introduce a series of regulations on service quality management and brand management. Guilin is a city with rich historical and cultural resources. Tourism management should strengthen the education and promotion of Guilin's history and culture among all tourism practitioners in the city, and improve their cultural literacy. Tourism management also needs to learn macro management methods from famous foreign tourist cities to improve the effectiveness and scientificity of tourism macro management.”

Thank you again for the revision suggestions from the reviewing experts. Your suggestion is an important guarantee for us to improve the paper and smoothly publish it. I look forward to continuing to exchange ideas in my future academic work and receiving your guidance.

---

## [Decision Letter · Decision Letter 1]

14 Dec 2023

A Study on the Impact of Tourism Destination Image and Local Attachment on the Revisit Intention: The Moderating Effect of Perceived Risk

PONE-D-23-31058R1

Dear Dr. Wei,

We’re pleased to inform you that your manuscript has been judged scientifically suitable for publication and will be formally accepted for publication once it meets all outstanding technical requirements.

Kind regards,

Bo Pu, Ph.D.

Academic Editor

PLOS ONE

Additional Editor Comments (optional):

this manuscript should be published in PLOS ONE.

Reviewers' comments:

Reviewer's Responses to Questions

**Comments to the Author**

1. If the authors have adequately addressed your comments raised in a previous round of review and you feel that this manuscript is now acceptable for publication, you may indicate that here to bypass the “Comments to the Author” section, enter your conflict of interest statement in the “Confidential to Editor” section, and submit your "Accept" recommendation.

Reviewer #3: All comments have been addressed

Reviewer #4: (No Response)

Reviewer #5: All comments have been addressed

2. Is the manuscript technically sound, and do the data support the conclusions?

Reviewer #3: Yes

Reviewer #4: Partly

Reviewer #5: Yes

3. Has the statistical analysis been performed appropriately and rigorously? 

Reviewer #3: Yes

Reviewer #4: I Don't Know

Reviewer #5: Yes

4. Have the authors made all data underlying the findings in their manuscript fully available?

Reviewer #3: Yes

Reviewer #4: No

Reviewer #5: Yes

5. Is the manuscript presented in an intelligible fashion and written in standard English?

Reviewer #3: Yes

Reviewer #4: No

Reviewer #5: Yes

6. Review Comments to the Author

Reviewer #3: I accept this paper to publish. The authors have made all the necessary changes previously suggested. The content of the study in this form meets expectations.

Reviewer #4: Dear authors,

thank you for the revision of your manuscript. However, I still do not find it publishable in a global journal. Namely:

a) The literature you introduced as an additional support to your hypotheses is still based on what seems to be your own research, or research coming from your close academic circle. Many of those literature items are coming from publishers or journals, which are sometimes regarded as belonging to lower-tier of the global academic literature. This shows either authors are not informed of the major previous research in the field, or insist on a selection of sources, which are not appropriate for publishing in a major journal.

b) It makes no sense to insist on your publication record in a reply to reviewers - the manuscript is considered on the basis on its own merit.

c) You have not addressed the language issues in the translation of the questionnaire items. In addition, the entire manuscript feels like being translated by using machine (software) translation, and needs a thorough copy-editing effort by an English language professional.

d) The sample size is not questionable, but rather the selection of a single Chinese tourist destination, which might be extremely attractive within China. However, the generalization of results could be (and is probably) affected by a range of factors, including cultural ones. This should be clearly discussed within the manuscript, and the related literature with related previous research should be cited and results should be compared to such extant literature. I

e) As previously indicated, I am not an expert on complex quantitative methods. As such, I was struggling a bit to understand your methods, and it seems you are using the CFA in AMOS to validate the model reliability, and then switch back to the SPSS and use the regression-based models to evaluate your moderation model. Bootstrapping then appears as a method of choice for testing mediation, which is fine. However, it is not clear how you performed it - results are just thrown at the reader. I miss coherence in the way you handle your results and don't remember ever seeing a paper, where authors seem to 'jump' across the methods and software tools, without any obvious reason to do so. As previously suggested, it would be expected that you do entire modelling in SPSS by using the AF Hayes PROCESS model. It has no specific data requirements and it is based entirely on regressions. You can evaluate regression models, based entirely on SPSS output. The other route would be to use AMOS (for CBA SEM) or Smart PLS (or any other comparable tool, if you choose to do the PLS SEM), if you wish to do CFA-based evaluation of your research model, followed by the CBA or PLS SEM evaluation of the moderation and mediation models. You can also choose to do both and compare results to obtain a more robust analysis (I've seen at least a few papers do that in my field, e.g. Haski-Leventhal, D., Pournader, M., & Leigh, J. S. (2020). Responsible management education as socialization: Business students’ values, attitudes and intentions. Journal of Business Ethics, 1-19.)

I wish you a lot of luck in the further revisions of this manuscript and your future work.

Reviewer #5: Dear Author/s,

My previous suggestions have mostly been revised. I thank you for your kind and sensitive response and revisions. There are spelling errors in some parts of the manuscript. I recommend that it be reconsidered.

Good luck.

7. PLOS authors have the option to publish the peer review history of their article (what does this mean?). If published, this will include your full peer review and any attached files.

Reviewer #3: No

Reviewer #4: No

Reviewer #5: No
